# Non-Coding RNAs as Biomarkers for Embryo Quality and Pregnancy Outcomes: A Systematic Review and Meta-Analysis

**DOI:** 10.3390/ijms24065751

**Published:** 2023-03-17

**Authors:** Wen Huang, Andy Chun Hang Chen, Ernest Hung Yu Ng, William Shu Biu Yeung, Yin Lau Lee

**Affiliations:** 1Department of Obstetrics and Gynaecology, School of Clinical Medicine, Li Ka Shing Faculty of Medicine, The University of Hong Kong, Hong Kong, China; 2Shenzhen Key Laboratory of Fertility Regulation, Reproductive Medicine Center, The University of Hong Kong—Shenzhen Hospital, Shenzhen 518000, China; 3Centre for Translational Stem Cell Biology, Building 17W, The Hong Kong Science and Technology Park, Hong Kong, China

**Keywords:** embryo quality, implantation, pregnancy outcomes, extracellular small non-coding RNAs, non-invasive biomarker

## Abstract

Despite advances in in vitro fertilization (IVF), there is still a lack of non-invasive and reliable biomarkers for selecting embryos with the highest developmental and implantation potential. Recently, small non-coding RNAs (sncRNAs) have been identified in biological fluids, and extracellular sncRNAs are explored as diagnostic biomarkers in the prediction of IVF outcomes. To determine the predictive role of sncRNAs in embryo quality and IVF outcomes, a systematic review and meta-analysis was performed. Articles were retrieved from PubMed, EMBASE, and Web of Science from 1990 to 31 July 2022. Eighteen studies that met the selection criteria were analyzed. In total, 22 and 47 different sncRNAs were found to be dysregulated in follicular fluid (FF) and embryo spent culture medium (SCM), respectively. *MiR-663b*, *miR-454* and *miR-320a* in FF and *miR-20a* in SCM showed consistent dysregulation in two different studies. The meta-analysis indicated the potential predictive performance of sncRNAs as non-invasive biomarkers, with a pooled area under curve (AUC) value of 0.81 (95% CI 0.78, 0.844), a sensitivity of 0.79 (95% CI 0.72, 0.85), a specificity of 0.67 (95% CI 0.52, 0.79) and a diagnostic odds ratio (DOR) of 8 (95% CI 5, 12). Significant heterogeneity was identified among studies in sensitivity (I2  =  46.11%) and specificity (I2  =  89.73%). This study demonstrates that sncRNAs may distinguish embryos with higher developmental and implantation potentials. They can be promising non-invasive biomarkers for embryo selection in ART. However, the significant heterogeneity among studies highlights the demand for prospective multicenter studies with optimized methods and adequate sample sizes in the future.

## 1. Introduction

Infertility is a prevalent health issue affecting millions of reproductive-aged couples worldwide. The use of in vitro fertilization (IVF) is increasing worldwide to help infertile couples achieve conception. Despite advances in technology, the pregnancy and live birth rates are around 35% and 25% per transfer, respectively [1,2]. Implantation failure and early pregnancy loss are common [3,4]. There is a need for effective selection of the best embryo for transfer, especially when a single embryo or blastocyst is being transferred [5].

The most commonly used non-invasive method of embryo selection is based on the morphological grading of embryos or blastocysts. Cleavage-stage embryos are assessed by the number and symmetry of blastomeres, the proportion of fragmentation, the presence of multinucleation and the compaction status [6]. The quality of the blastocyst is determined by the degree of blastocoel expansion and the appearance of the inner cell mass and the trophectoderm [7]. However, the morphological assessment method is subjective, with inter- and intra-assessor variability [8,9,10]. Thus, morphological grading alone exhibits a poor correlation with the success rate of IVF [11]. Another non-invasive method for embryo selection is based on morphokinetic algorithms evaluated by time-lapse technology. Although time-lapse technology has been shown to improve embryo selection [12,13,14], a recent meta-analysis indicated that time-lapse systems made no significant improvement to pregnancy outcomes when compared with the conventional incubation of embryos, challenging the advantages of time-lapse technology in embryo culture and selection [15]. Notably, preimplantation genetic testing for aneuploidies (PGT-A) has been increasingly applied to distinguish aneuploid from euploid embryos [16]. However, a recent meta-analysis revealed that preimplantation genetic testing for aneuploidies performed on blastocysts only improved clinical outcomes in women over 35 but was ineffective in the general population [17]. Moreover, preimplantation genetic testing for aneuploidies requires the invasive biopsy of blastocysts, which may affect embryo developmental potential [18]. Therefore, there is a high demand for new non-invasive and easy-to-perform methods of embryo selection.

Recently, the discovery of cell-free DNA in SCM prompted the development of non-invasive PGT-A (niPGT-A) [19]. Although niPGT-A has emerged as a potential alternative to conventional PGT-A for safety and economic reasons, the contamination of maternal cell-free DNA in SCM and the concordance between PGT-A and niPGT-A still need to be determined [20]. Various biomarkers for the non-invasive assessment of embryo viability and implantation potential have been proposed. Some studies have used the metabolomic and proteomic profiling of spent embryo culture medium (SCM) in combination with time-lapse technology to predict implantation rates [21,22,23]. These methods rely on highly sensitive instruments, restricting their widespread use. Emerging attention has been paid to the use of small non-coding RNAs (sncRNAs) in predicting embryo implantation ability. The sncRNAs are highly abundant RNAs with sizes of typically <100 nucleotides long, and they play essential roles in diverse biological processes, including embryogenesis and embryo implantation [24,25,26]. The main categories of sncRNAs include microRNAs (miRNAs), long non-coding RNAs (lncRNAs), P-element-induced-wimpy-testis-(PIWI)-interacting RNAs (piRNAs) and circular RNAs (circRNAs) [27]. Apart from expressing within the cells, sncRNAs can also be detected in extracellular fluids such as embryo SCM [28], follicular fluid (FF) [29], uterine fluid [30], serum [31] and seminal plasma [32]. More importantly, sncRNAs are highly stable in extracellular environments. As compared to whole genome DNA sequencing for niPGT-A, the highly sensitive and fast detection of a panel of sncRNA markers using real-time quantitative PCR techniques [33] may offer an advantage for embryo evaluation in ART practice.

To comprehensively understand the potential of using sncRNAs as non-invasive biomarkers in embryo assessment, we systematically reviewed studies investigating the correlation of sncRNA levels in various extracellular environments with pregnancy outcomes or embryo quality parameters. We also evaluated the predictive accuracy of cell-free sncRNAs from the included studies as non-invasive biomarkers to predict implantation outcomes.

## 2. Materials and Methods

### 2.1. Data Sources and Search Strategy

The meta-analysis was conducted according to the guidelines of the Preferred Reporting Items for Systematic Reviews and Meta-Analyses (PRISMA, Berlin, Germany) [34,35]. The review protocol was registered with PROSPERO (CRD42022350657) on 12 August 2022 before performing the preliminary research. Three databases, PubMed, Web of Science and EMBASE were searched to identify the related literature. Keywords, including “non-coding RNAs”, “microRNAs”, “untranslated RNA”, “embryo implantation” and “embryo quality”, were searched as medical subject headings (MeSH), Emtree headings or free-text terms. The search strategies for all databases are listed in Appendix A.

### 2.2. Inclusion and Exclusion Criteria

The inclusion and exclusion criteria were defined using the PICOS (population, intervention, comparator, outcome, study) approach [36]. The details are summarized in Appendix A. Only human studies, and not animal or in vitro studies, were included. As the systemic review focused on the correlation of sncRNAs in extracellular fluids with embryo quality or pregnancy outcomes, reports detecting sncRNA abundance within cells or tissues were excluded. Regarding the study design, comparative studies, cross-sectional studies and retrospective and prospective studies from peer-reviewed journals were included, whereas review articles, letters, commentary articles and conference abstracts were excluded. Furthermore, only publications in English with dates ranging from 1990 to 31 July 2022 were included.

### 2.3. Study Selection and Quality Assessment

Results obtained from PubMed, EMBASE and Web of Science were imported into EndNote 20. All records contained the following information: title, authors, publication year, digital object identifier (DOI), accession number and abstract. All information was first screened by one author (W.H.) for eligibility and cross-checked by another author (A.C.). The references identified as potentially eligible were then fully assessed against the inclusion and exclusion criteria. Any discrepancies between individuals were re-evaluated by the other two authors (W.Y. and C.L.). Since all included articles were observational studies, the quality was assessed using the standard scale “Quality assessment tool for observational cohort and cross-sectional studies” from the NHLBI (https://www.nhlbi.nih.gov/health-topics/study-quality-assessment-tools (accessed on 17 August 2022) by two authors (W.H. and A.C.) independently. Disagreements between the review authors over the risk of bias in particular studies were resolved by the other two authors (W.Y. and C.L.) referencing the original article. After evaluation, the qualities of all the included studies ranged from fair to good.

### 2.4. Data Extraction

After confirming the inclusion of the articles, the following information was extracted from the full text for analysis: title, authors, year of publication, journal, study design, participant characteristics, sample type, the detection method of sncRNAs, sncRNA expression levels, and their correlation with pregnancy outcomes or embryo quality.

### 2.5. Statistical Analysis

The area under the curve (AUC) value, the 95% confidence interval (CI) and the sensitivity and specificity of individual sncRNA, if provided in the study, were extracted. The values of true positive (TP), false positive (FP), false negative (FN) and true negative (TN) were calculated according to the following formulas: Sensitivity = Number of TP/Total number of individuals with positive outcomes (successful implantation or good quality embryos), and Specificity = Number of TN/Total number of individuals with negative outcomes (failed implantation or poor-quality embryos). The pooled diagnostic value, sensitivity, specificity, positive likelihood ratio (PLR), negative likelihood ratio (NLR), diagnostic odds ratio (DOR), and corresponding 95% CI of sncRNAs were analyzed using Stata SE 15.1 (StataCorp LLC, College Station, TX, USA). Publication bias was examined by Deeks’ funnel plot, with values showing *p* < 0.1 considered to be significant. We applied the chi-squared-based Cochran Q test and Higgin’s I2 statistics to assess heterogeneity among the studies. Significant heterogeneity was defined when *p* < 0.10 or I2 > 50%.

## 3. Results

### 3.1. Identification and Selection of Studies

After the initial search, 134 articles, including 52 from PubMed, 35 from EMBASE and 47 from Web of Science, were recorded (Figure 1). After removing duplicated records among the three databases, a total of 95 records were subjected to title and abstract screenings. Among them, 70 did not meet the inclusion criteria. The full texts of the remaining 25 articles were downloaded for further assessments on the eligibility; 7 were excluded due to the following reasons: not relevant comparators (*n* = 2); studies with additional intervention (*n* = 2); descriptive studies without comparison (*n* = 1); not detecting specific sncRNAs (*n* = 2). Finally, 18 studies were confirmed eligible for this systematic review according to the selection criteria.

### 3.2. Characteristics of the Reviewed Studies

The baseline characteristics of the selected studies are summarized in Table 1. The 18 included articles were all published within the past ten years. There were 12 and 6 studies using SCM and FF, respectively, as non-invasive samples to investigate the differentially expressed sncRNAs. The average age of women recruited in the studies ranged from 27.9 to 36.3 years. The causes of infertility indicated in the studies were mainly male or tubal factors. While seven studies used ICSI alone and three studies used IVF alone as the insemination method, seven other studies used both methods. One of the articles did not mention the insemination method at all [37]. Embryo morphological assessment methods were specified in 12 studies: Gardner grading scale in 4 studies, ALPHA/ESHRE guidelines in 4 studies, Veeck’s criteria in 2 studies, and Tao’s criteria and the Eeva system in 1 study.

### 3.3. Identification of Differentially Expressed sncRNAs in Follicular Fluid

Out of the six studies which identified sncRNA from FF, three used the antagonist protocol [46,48,49], one used the agonist protocol [40] and the remaining two included patients who received either the antagonist or agonist protocol [45,59] for ovarian stimulations. Taken together, the expressions of 22 miRNAs were associated with embryo quality, development potential and/or pregnancy outcomes. It was found that *miR-663b* [46,49] and the *miR-320* family [40,59] were upregulated in good-quality embryos at day 3 and in blastocysts, while *miR-454* [46,48] was downregulated in two different studies (Table 2).

### 3.4. Identification of Differentially Expressed sncRNAs in Embryo Spent Culture Medium

SCM samples were collected on day 3, day 4 and/or day 5 after oocyte retrieval for sncRNA profiling and validation analyses in 12 studies. Of these, 10 studies compared differential sncRNA expression between non-pregnant and pregnant groups after embryo transfer. In four studies, different embryo qualities defined by various criteria were adopted as comparators. Overall, 24 miRNAs and 23 piRNAs were associated with embryo quality or pregnancy outcomes. Only the cell-free miRNA-20a family [42,51,57] was found to be highly concentrated in good-quality or successfully implanted embryos in three studies. However, contradictory results were found in the correlation of four miRNAs, namely, miR-30c [42,57], miR-21-5p [51,54], miR-191 [38,56] and let-7i-5p [50,52,55], with pregnancy outcomes (Table 3).

### 3.5. Predictive Ability of Differentially Expressed sncRNAs

The AUC value of the receiver operating characteristic (ROC) curve is commonly used for assessing the discriminative ability of a prediction model. The higher the AUC (near to 1), the better the model’s performance at distinguishing between two different classes [60]. The statistical analysis of different sncRNAs, including AUC value, 95% CI, sensitivity, and specificity, if provided in the study, is summarized in Table 4. Overall, the predictive ability of 10 miRNAs and one combination of 4 sncRNAs was assessed for pregnancy outcomes, day 3 embryo quality, blastocyst formation or expanded blastocyst. Of these, the AUC values of three miRNAs, miR-19b-3p, miR-15a-5p and miR-20a-5p, in SCM were higher than 0.8, which is considered excellent, while the other seven miRNAs and the combination of sncRNAs had acceptable AUC values ranging from 0.6 to 0.7 [61]. In SCM, miR-20a and miR-20a-5p could predict pregnancy outcomes and day 3 embryo quality with AUC values of 0.773 (0.737–0.908) and 0.855 (0.746–0.965), respectively. MiR-320a in SCM [51] and FF [59] had similar prediction values for day 3 embryo quality, which were 0.768 (0.633–0.904) and 0.753 (0.651–0.855), respectively. Moreover, the AUC values of miR-21-5p were 0.736 (0.639–0.833) [54] in SCM and 0.774 (0.628–0.856) [59] in FF for the prediction of pregnancy outcomes and 0.753 (0.609–0.898) [51] in SCM for the prediction of day 3 embryo quality. 

### 3.6. Predictive Efficacy of Extracellular sncRNAs for Embryo Development Potential and Pregnancy Outcomes

To evaluate the performance of sncRNAs in predicting IVF outcomes, we further performed a meta-analysis on three studies [45,55,59] with raw statistics available. As shown in Figure 2, the pooled sensitivity and specificity were 0.79 (95% CI 0.72, 0.85) and 0.67 (95% CI 0.52, 0.79), respectively. Moreover, there were significant heterogeneities among studies in sensitivity (I2  =  46.11%) and specificity (I2  =  89.73%) (*p* <  0.1) (Figure 2A,B). The other pooled parameters determined were AUC 0.81 (95% CI 0.78, 0.844); PLR 2.39 (95% CI 1.6, 3.4); NLR 0.31 (95% CI 0.25, 0.40) (Figure 2C). The PLR of 2.39 showed that an embryo with successful implantation was about 2.4 times more likely to have a positive test result than an embryo that failed to implant. Deeks’ funnel plot asymmetry test was adopted, and a *p*-value of 0.53 indicated the absence of publication bias in the current meta-analysis (Figure 2D). Furthermore, the DOR value was 8 (95% CI 5, 12), which signified that embryos that had tested positive for extracellular sncRNA had an 8-fold higher chance of successful implantation than those that tested negative (Figure 2D). Given that the prevalence of successful pregnancy per embryo transfer was around 35% [1,2], the pretest probability was defined at 35%. Fagan’s plot demonstrated that the post-test probability of successful implantation for a positive test result increased from 35% to 56% and dropped from 35% to 14% with a negative test result (Figure 2E). Taken together, these results indicated that extracellular sncRNAs might serve as a good predictive index for successful implantation with relatively high accuracy.

## 4. Discussion

In the past decade, a number of studies explored the potential use of sncRNAs in SCM or FF as non-invasive biomarkers for assessing embryo quality and predicting pregnancy outcomes. However, the discrepancy of sncRNA biomarkers identified in various studies prompted us to systematically review the available literature about the differentially expressed extracellular sncRNAs and their potential roles in embryonic development and implantation. The studies included in this systemic review and meta-analysis showed that in vitro cultured embryos secreted different sncRNA profiles into SCM and that some sncRNAs were associated with implantation outcomes following IVF. Embryo quality and pregnancy outcomes were also related to sncRNA contents in the FF. Overall, three miRNAs (*miR-663b*, *miR-454* and *miR-320*) in FF and one miRNA (*miR-20a*) in SCM were consistently found to be dysregulated in more than one study. Our meta-analysis showed the predictive performance of sncRNAs as non-invasive biomarkers, with a pooled AUC value of 0.81, sensitivity of 0.79, specificity of 0.67, PLR of 2.4 and NLR of 0.31, indicating the satisfactory predictive accuracy [61] of extracellular sncRNAs in embryo quality and pregnancy outcomes.

It is known that oocyte quality is crucial to embryo development potential [62]. The oocyte is surrounded by FF, which is a complex and dynamic microenvironment providing nourishment and regulatory molecules for oocyte development and maturation. Therefore, FF is a critical determinant of oocyte competency [63]. The identification of miRNAs in human FF as predictors of oocyte and embryo quality and pregnancy outcomes has been reported [59,64]. Our analysis demonstrates that the upregulation of *miR-320* [40,59] and *miR-663b* [46,49] and the downregulation of *miR-454* [46,48] in FF have been consistently reported as being associated with good-quality embryos in more than one study. Microinjection of the *miR-320a* inhibitor into mouse oocytes significantly impaired oocyte competency and subsequent embryo development by inhibiting Wnt signaling [40]. Oxidative stress can initiate oocyte aging [65] and increase apoptosis in human granulosa cells (GCs), which surround and nourish oocytes [66]. In agreement, *miR-320a* reduces reactive oxygen species levels by targeting *Sirtuin 4* (*SIRT4*) in oocytes and GCs [67]. Moreover, *miR-320a* potentiated ovarian steroidogenesis in GCs through modulation of *CYP11A1* and *CYP19A1* expression by directly targeting the osteogenic transcription factor *RUNX2* [68]. These results indicate the crucial roles of *miR-320a* in oocyte maturation and development. Although *miR-663b* [69,70] and *miR-454* [71,72] have been extensively studied in various cancers, their regulatory roles in oocyte competence and embryo development are unknown. 

Interestingly, the upregulation of *miR-21-5p* in FF [59] and its downregulation in SCM [54] are associated with successful pregnancies. Moreover, *miR-21-5p* in FF and SCM is predictive of pregnancy outcomes with relatively high AUC values [54,59]. *MiR-21-5p* is enriched in cumulus cells associated with oocytes that develop into high-quality blastocysts [73]. Inhibition of *miR-21-5p* in cumulus cells induces cell apoptosis via the upregulation of *phosphatase and tensin homolog* (*PTEN*) [73]. While a significant increase of *miR-21-5p* was found from 1- to 8-cell stage bovine embryos, its level was downregulated at the blastocyst stage, leading to the expression of the *miR-21-5p*-suppressed genes required for blastocyst formation [74]. It may explain the contradictory findings that *miR-21-5p* was upregulated in good quality 8-cell stage embryos [51] while it was downregulated in blastocysts with successful implantation [54]. In addition, high-throughput miRNA sequencing revealed that the miRNA profiles were distinct between the cleavage and blastocyst SCM [54], suggesting that the developmental stages of the embryo could affect the sncRNA profiles in SCM.

*MiR-20a* in SCM is predictive of pregnancies with relatively high AUC values [42]. Consistently, miR-20a expression is positively correlated with embryo quality [51,57]. As one of the members in the *miR-17-92* cluster [75], *miR-20a* is detectable in both inner cell mass and the trophectoderm of mouse blastocysts. It silences signal transducer and activator of transcription 3 (*Stat3*), a downstream transcription factor in the JAK-STAT signal cascade [76]. Considering the critical role of STAT3 in sustaining the self-renewal of both mouse [77] and human [78] embryonic stem cells (ESC), its negative regulation by *miR-20a* might be associated with the onset of differentiation in ESC and human embryos [76]. Another target of *miR-20a*, *bone morphogenic protein receptor 2* (*Bmpr2*) [79], could phosphorylate and activate the Smad family in the presence of BMP ligands, thereby promoting the expression of the inhibitor of *differentiation genes 1 and 3* (*Id1/3*) [80] and the *gap junction protein connexin 43* (*CX43*) [81]. The downregulation of *Bmpr2* and its downstream genes by *miR-20a* initiates cell differentiation in ESC and gastrulation in embryos [79]. Moreover, *miR-20a* could rescue the proliferation defect in Dgcr8-knockout ESC by promoting the G1-S transition of the cell cycle [82]. These findings suggest that high levels of *miR-20a* in SCM may indicate the differentiation potential of good-quality embryos.

Three members of *let-7*, including *let-7a-5p* [55], *let-7b-5p* [52] and *let-7i-5p* [52,55], were dysregulated in the SCM of poor-quality and non-implanted embryos. The *let-7* family is one of the first miRNA families discovered in *C. elegans*, and it consists of 12 members in humans [83]. *Lin-28* inhibits the biogenesis of *let-7*, which, in turn, suppresses *lin-28* expression [84]. *Lin-28* is critical to the self-renewal of pluripotent stem cells [85], while a high level of *let-7* is associated with induced cell differentiation and reduced cell proliferation [85,86]. Interestingly, a combination of *let-7b-5p* and *let-7i-5p*, together with two piRNAs (*piR-016735* and *piR-020381*), can predict pregnancy outcomes modestly [55]. Therefore, it is also worth evaluating the practicability of combined extracellular snRNAs in predicting pregnancy outcomes in future research.

Chromosomal abnormality contributes to implantation failure and miscarriages in IVF [87]. PGT-A has been widely used to distinguish aneuploid from euploid embryos [16]. However, a biopsy of only 5 to 10 trophectoderm cells cannot unequivocally represent the whole embryo, especially for mosaic embryos [88]. PGT-A results alone may not accurately predict the ultimate chromosomal fate of the resulting fetus [89], and its effectiveness in improving pregnancy outcomes is still questionable [17]. Interestingly, it has been reported that higher *miR-191* was found in the SCM of aneuploid embryos and was negatively correlated with successful implantation [38]. These findings suggest that extracellular sncRNAs might reflect chromosomal abnormalities in embryos and could be considered an additional metric for embryo selection.

The association of differential expressions of piRNAs in SCM with embryo quality and pregnancy outcomes was reported [50,52,55]. PiRNAs are a class of sncRNAs (23–30 nucleotide) that interact with the PIWI proteins of the Argonaute/PIWI family. The first described function of PIWI–piRNAs complexes is the silencing of transposable elements, thereby maintaining genomic integrity [90]. Gametes and early embryos are susceptible to transposable element reactivation during genome reprogramming [91]. The PIWI–piRNAs are crucial for mammalian gametogenesis and embryogenesis [92]. Deletion of the piRNA pathway proteins results in the upregulation of transposable elements and considerable transcriptomic dysregulation during spermatogenesis and oogenesis in various animal models [93,94,95]. In humans, mutations of the PIWI pathway genes were observed in infertile men [96,97]. According to the data reviewed in the present study, a variety of piRNAs were dysregulated in the SCM of non-implanted or poor-quality embryos [50,52,55]. In silico analysis further revealed that the predicted target genes (e.g., *TEAD3*, *ELF1*, *SP3*, *GBX2*, etc.) of the differentially expressed piRNAs were involved in gamete maturation and preimplantation development [50,52,55], suggesting the important roles of piRNAs in embryogenesis. There is no report on the presence of piRNAs in FF, despite the presence of PIWI-like RNA-mediated gene silencing 3 (PIWIL3) and piRNAs in human oocytes [98].

Embryo-derived sncRNAs are involved in blastocyst–endometrium communication and are indispensable for successful embryo implantation. Indeed, miRNAs are absent in culture media derived from cleavage and morula stage embryos, indicating miRNA profiling from SCM is blastocyst-specific [42]. In addition, the expression profile of sncRNAs in SCM was distinct from that of the trophectoderm and inner cell mass [99], suggesting the unique roles of secretory embryonic sncRNAs in the interaction with the endometrium. SCM from non-implanted embryos impairs endometrial receptivity by modulating the expressions of a number of genes (e.g., *VEGF-A*, *HOXA10* and *TGF-β1*) in primary human endometrial epithelial cells (HEECs) [100,101]. Coincidentally, miR-661 from the non-implanted blastocysts is taken up by primary HEECs and inhibits endometrial cell adhesiveness via targeting *poliovirus receptor-related 1* (*Pvrl1*) [39]. These studies support the critical roles of embryo-derived sncRNAs in regulating endometrial receptivity and, thereby, affecting the outcomes of pregnancy.

The current systematic review and meta-analysis have several limitations. First, due to the small number of studies and significant heterogeneity in the embryo quality assessment and pregnancy outcome parameters, subgroup analysis was not performed in the meta-analysis. Various embryo grading systems were applied to evaluate embryo quality in the reviewed studies, such as the Gardner grading system, Veeck’s criteria and the ALPHA/ESHRE guidelines. In addition, two types of ovarian stimulation protocols, including antagonist and agonist protocol, were used to obtain a higher number of oocytes. However, the type and dose of gonadotropin, a critical drug in ovarian stimulation protocols, were found to affect the miRNA expression levels in FF [59]. Thus, the embryo evaluation standards and ovarian stimulation protocols may contribute to the heterogeneity. Second, some included studies targeted specific sncRNAs rather than global sncRNA profiling, which might have resulted in selection bias. Finally, and most importantly, there are no standardized protocols or techniques for sncRNA-analysis in SCM. The methodologies, including sncRNA isolation and quantification, cDNA library construction and the types of reference genes for data normalization, which might lead to result variation, should be optimized and standardized [102,103]. Therefore, the predictive value of extracellular sncRNAs in embryo assessment should be interpreted with caution and confirmed in large-scale studies. Nevertheless, this is the first systematic review and meta-analysis trying to identify extracellular sncRNAs for predicting embryo quality and pregnancy outcomes.

## 5. Conclusions

In summary, we have shown that extracellular sncRNAs such as *miR-320*, *miR-20a* and *miR-21-5p* are potential non-invasive biomarkers to evaluate the developmental potential of embryos. The association of the dysregulation of extracellular sncRNAs with pregnancy outcomes indicates the importance of sncRNAs in regulating embryo development and embryo–endometrial interaction. However, significant heterogeneity was observed among studies, likely due to the differences in study design and the detection methods of sncRNAs. Further multicenter and prospective studies with standardized methods should be conducted to investigate the clinical value of sncRNAs as biomarkers for embryo quality and pregnancy outcomes.

## Figures and Tables

**Figure 1 ijms-24-05751-f001:**
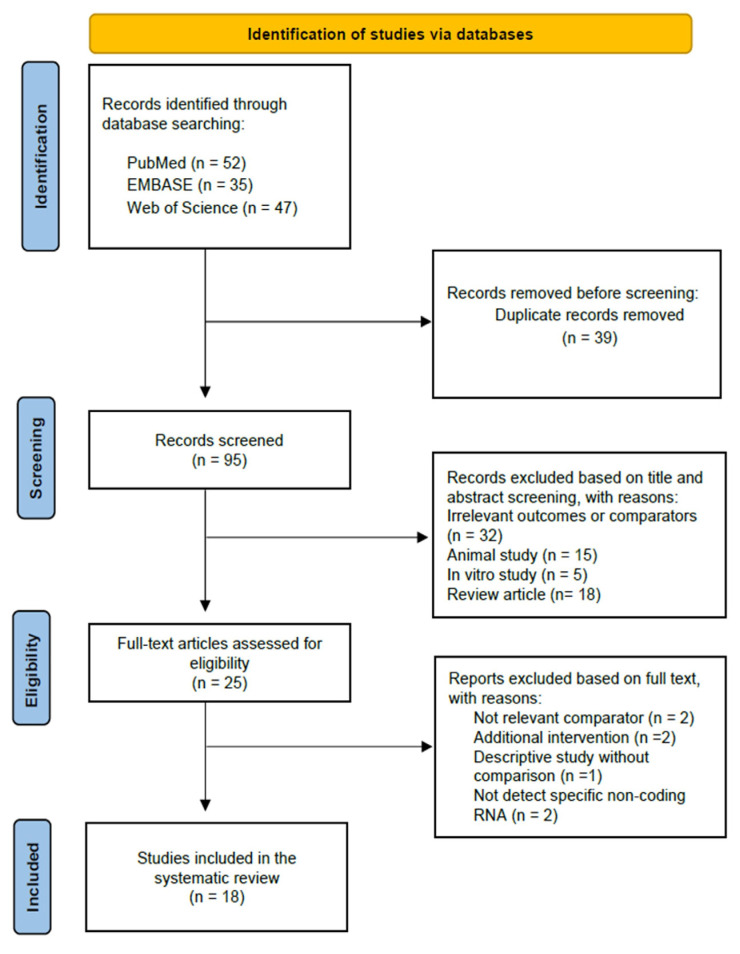
Flowchart of the literature search and selection process.

**Figure 2 ijms-24-05751-f002:**
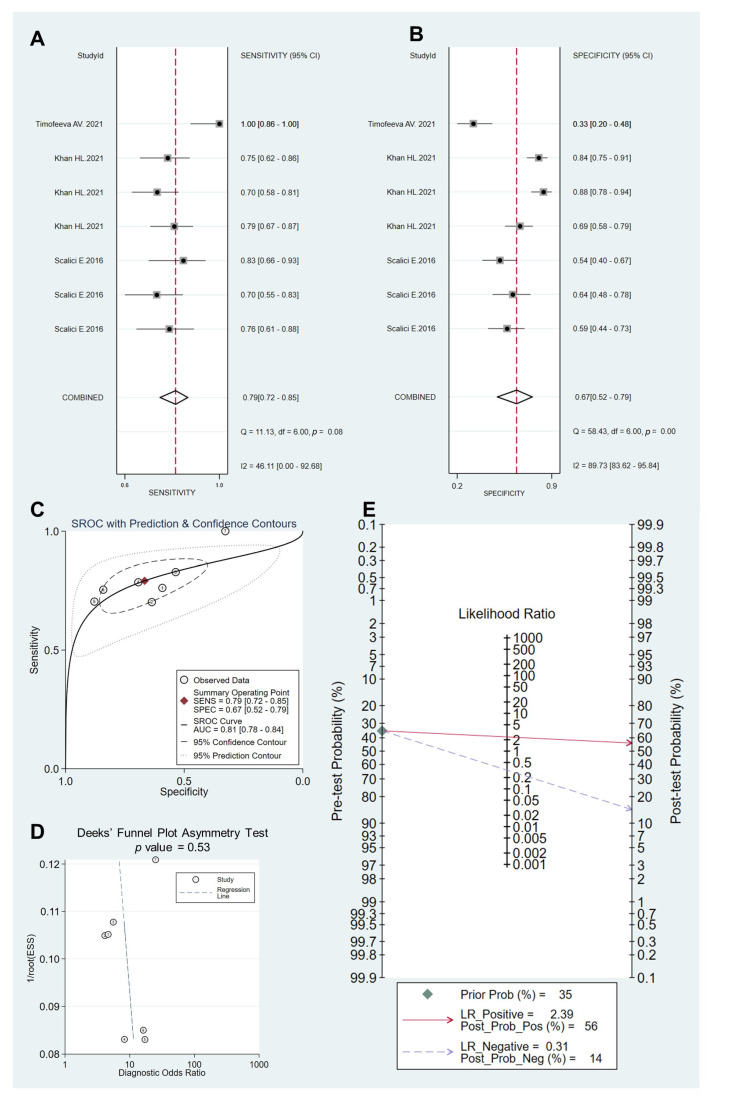
Pooled predictive performance of extracellular sncRNAs for embryo implantation potential. Data from three published articles [45,55,59] were analyzed. (**A**) Forest plots of the pooled sensitivity. (**B**) Forest plots of the pooled specificity. (**C**) SROC curve. (**D**) Deeks’ funnel plot. (**E**) Fagan’s plot.

**Table 1 ijms-24-05751-t001:** Characteristics of reviewed studies.

Reference (First Author)	Sample Type	Mean Age (Years)	Causes of Infertility	Insemination Method	Embryo Assessment Method	Pregnancy Diagnosis Method
Rosenbluth EM. 2014 [38]	SCM	NA	NA	IVF/ICSI	NA	Live birth
Cuman C. 2015 [39]	SCM	34.8	Male factors	ICSI	NA	Term pregnancy
Feng R. 2015 [40]	FF	34.3	Male factors, Tubal factors	ICSI	Veeck L. [41]	NA
Capalbo A. 2016 [42]	SCM	NA	NA	IVF	Gardner DK. [43]	Fetal heartbeat
Borges E. 2016 [44]	SCM	NA	NA	ICSI	NA	Implantation *
Scalici E. 2016 [45]	FF	34.3	Primary (*n* = 57) or Secondary infertility (*n* = 34)	IVF/ICSI	Gardner DK. [7]	Fetal heartbeat
Machtinger R. 2017 [46]	FF	30.7	Male factors, Mechanical or Unexplained factors	IVF/ICSI	ALPHA/ESHRE [47]	NA
Martinez RM. 2018 [48]	FF	31.0	NA	IVF/ICSI	ALPHA/ESHRE [47]	NA
Fu J. 2018 [49]	FF	30.5	Primary (*n* = 64) or Secondary infertility (*n* = 27)	ICSI	Gardner DK. [7]	NA
Timofeeva AV. 2019 [50]	SCM	32.0	Tubal factors, Male factors	IVF/ICSI	Gardner DK. [43]	β-HCG level
Abu-Halima M. 2020 [51]	SCM	27.9	NA	ICSI	Veeck L. [41]	Implantation *
Timofeeva AV. 2020 [52]	SCM	33.0	Male factors, Tubal factors, DOR	ICSI	Tao J. [53]	Live birth
Fang F. 2021 [54]	SCM	30.7	Primary (*n* = 43) or Secondary infertility (*n* = 17)	IVF/ICSI	ALPHA/ESHRE [47]	Implantation *
Wang S. 2021 [37]	SCM	30.3	NA	NA	ALPHA/ESHRE [47]	Fetal heartbeat
Timofeeva AV. 2021 [55]	SCM	32.4	Primary (*n* = 52) or Secondary infertility (*n* = 58)	ICSI	NA	β-HCG level
Acuña-González RJ. 2021 [56]	SCM	36.3	NA	IVF	NA	Gestational sac
Coticchio G. 2021 [57]	SCM	36.0	Male factors, Tubal factors, Polycystic ovary	IVF/ICSI	Eeva system [58]	NA
Khan HL. 2021 [59]	FF	34.6	Female (*n* = 48) or Unexplained factors (*n* = 97)	IVF	ALPHA/ESHRE [47]	Fetal heartbeat

*: Details not provided. SCM: embryo spent culture medium; FF: follicular fluid; ART: assisted reproductive technology; IVF: standard in vitro fertilization; ICSI: intracytoplasmic sperm injection; DOR: diminished ovarian reserve; Eeva system: early embryo viability assessment system; ESHRE: European society of human reproduction and embryology; NA: data not available.

**Table 2 ijms-24-05751-t002:** List of differentially expressed sncRNAs in the follicular fluid after validation by qRT-PCR in the included studies.

Dysregulated sncRNAs	Comparison Groups	Stimulation Protocol	Ref.
Down: miR-320, miR-197	Poor- (*n* = 24) vs. Good-quality (*n* = 29) day 3 embryo	Agonist protocol (*n* = 53)	[40]
Up: let-7b	No blastocyst vs. Viable blastocyst	Antagonist protocol (*n* = 48) Agonist protocol (*n* = 39)	[45]
Up: let-7b	Non-expanded vs. Expanded blastocyst		
Down: miR-29a	Non-pregnant vs. Pregnant		
Down: miR-202-5p, miR-206, miR-16-1-3p, miR-1244	Failed (*n* = 5) vs. Normal fertilization (*n* = 30)	Antagonist protocol	[46]
Up: miR-454-5p, miR-425-3p, miR-16-5p, miR-222-3p	Abnormal (*n* = 4) vs. Normal fertilization (*n* = 30)		
Down: miR-766-3p, miR-663b, miR-132-3p, miR-16-5	Impaired (*n* = 10) vs. Top-quality (*n* = 19) day3 embryo		
Down: miR-92a, miR-130b	Failed (*n* = 33) vs. Normal fertilization (*n* = 93)	Antagonist protocol	[48]
Down: miR-888. Up: miR-214, miR-454	Impaired (*n* = 48) vs. Top-quality (*n* = 42) day 3 embryo		
Down: miR-663b	No blastocyst (*n* = 53) vs. Viable blastocyst (*n* = 38)	Antagonist protocol	[49]
Down: miR-663b	Poor-scoring (*n* = 21) vs. Top-scoring (*n* = 17) blastocyst		
Down: miR-320a	Non-top-quality embryo vs. Top-quality day 3 embryo	Antagonist protocol (*n* = 73) Agonist protocol (*n* = 72)	[59]
Down: miR-212-3p	No blastocyst vs. Viable blastocyst		
Down: miR-212-3p	Non-expanded blastocyst vs. Expanded blastocyst		
Down: miR-21-5p	Non-pregnant vs. Pregnant		

Up: upregulated sncRNA in the former group compared to the latter group; Down: downregulated sncRNAs in the former group compared to the latter group. Ref: reference.

**Table 3 ijms-24-05751-t003:** Summary of differentially expressed sncRNAs in spent culture medium after validation by qRT-PCR in the included studies.

Dysregulated sncRNAs	Comparison Groups	Collection Time	Ref.
Up: miR-191	Aneuploid (*n* = 19) vs. Euploid (*n* = 9) embryos	Day 4 and 5	[38]
Up: miR-645, miR-372, miR-191	Non-pregnant (*n* = 9) vs. Pregnant (*n* = 18)		
Up: miR-661	Non-pregnant (*n* = 13) vs. Pregnant (*n* = 13)	Day 5	[39]
Down: miR-20a, miR-30c	Non-pregnant (*n* = 28) vs. Pregnant (*n* = 25)	Day 3 to5	[42]
Up: miR-142-3p	Non-pregnant (*n* = 18) vs. Pregnant (*n* = 18)	Day 3	[44]
Up: let-7i-5p	Poor (n = 6) vs. Excellent (*n* = 32)	Day 4	[50]
Down: piR-17716	Poor (n = 6) vs. Good (*n* = 16)		
Down: piR-16735	Fair (n = 11) vs. Good (*n* = 16)		
Up: piR-020401, let-7i-5p	Non-pregnant (*n* = 25) vs. Pregnant (*n* = 14)		
Up: miR-320a, miR-15a-5p	G2 (*n* = 23) vs. G1 (*n* = 23) *	Day 3	[51]
Down: miR-21-5p	G3 (*n* = 23) vs. G1 (*n* = 23) *		
Down: miR-423-5p, miR-20a-5p	G3 (*n* = 23) vs. G2 (*n* = 23) *		
Up: miR-19b-3p	Non-pregnant (*n* = 22) vs. Pregnant (*n* = 24)		
Down: piR-011291, piR-019122, piR-001311, piR-015026, piR-015462, piR-016735, piR-019675, piR-020381, piR-020485, piR-004880, piR-000807, let-7b-5p, let-7i-5p	Morula without (*n* = 20) vs. with (*n* = 29) blastulation potential	Day 4	[52]
Up: miR-26b-5p, miR-21-5p	Non-pregnant (*n* = 30) vs. Pregnant (*n* = 30)	Day 3 and 5	[54]
Up: miR-483-5p (Day3), miR-432-5p (Day5); Down: miR-199a-3p > miR-199b-3p, miR-199a-5p, miR-379-5p, miR-99a-5p (Day 5)	Non-pregnant (*n* = 3) vs. Pregnant (*n* = 5)	Day 3 and 5	[37]
Up: piR-020485, piR-015249 (Day4); Down: piR000765, piR-022628, let-7i-5p, piR-008112, piR-022258, piR-015026 (Day4), piR-008113, miR-381-3p, let-7a-5p, piR-001312 (Day5)	Non-pregnant (*n* = 49) vs. Pregnant (*n* = 25)	Day 4 and 5	[55]
Up: miR-24-1-5p; Down: miR-191-5p	Non-pregnant (*n* = 25) vs. Pregnant (*n* = 25)	Day 5	[56]
Up: miR-30c; Down: miR-20a	Eeva scores (from 5 to 1) (*n* = 136)	Day 5	[57]

* Embryo grading based on the criteria of Veeck. G1 embryos having excellent quality and G5 embryos having the poorest quality. Up: upregulated sncRNA in the former group compared to the latter group. Down: downregulated sncRNAs in the former group compared to the latter group. Ref: reference.

**Table 4 ijms-24-05751-t004:** Summary of predictive ability of differentially expressed sncRNAs in various studies.

Prediction	sncRNAs	AUC	95% CI	Sensitivity (%)	Specificity (%)	Sample Type	Ref.
Pregnancy outcome	miR-20a	0.773	0.737–0.908	NA	NA	SCM	[57]
	miR-30c	0.786	0.663–0.909	NA	NA	SCM	[57]
	miR-19b-3p	0.818	0.696–0.940	NA	NA	SCM	[51]
	miR-26b-5p	0.725	0.622–0.829	NA	NA	SCM	[54]
	miR-21-5p	0.736	0.639–0.833	NA	NA	SCM	[54]
	piR-016735 + piR-02038+ let-7b-5p + let-7i-5p	0.708	NA	100	33.3	SCM	[52]
	miR-29a	0.680	0.550–0.790	83.3	53.5	FF	[45]
	miR-21-5p	0.774	0.628–0.856	74.8	83.7	FF	[59]
Day 3 embryo quality	miR-320a	0.768	0.633–0.904	NA	NA	SCM	[51]
	miR-15a-5p	0.815	0.691–0.937	NA	NA	SCM	[51]
	miR-21-5p	0.753	0.609–0.898	NA	NA	SCM	[51]
	miR-20a-5p	0.855	0.746–0.965	NA	NA	SCM	[51]
	miR-320a	0.753	0.651–0.855	80.0	71.0	FF	[59]
Blastocyst formation	let-7b	0.660	0.550–0.760	77.2	59.1	FF	[45]
	miR-212-3p	0.744	0.648–0.841	79.0	69.0	FF	[59]
Expanded blastocyst	let-7b	0.670	0.540–0.790	70.0	64.3	FF	[45]
	miR-212-3p	0.726	0.623–0.829	71.0	88.0	FF	[59]

SCM: embryo spent culture medium; FF: follicular fluid; 95% CI: 95% confidence interval; NA: data not available. Ref: reference.

## Data Availability

Data will be made available upon reasonable request.

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
