# Peer review of "Non-Coding RNAs as Biomarkers for Embryo Quality and Pregnancy Outcomes: A Systematic Review and Meta-Analysis"

_ijms, 2023, doi:10.3390/ijms24065751_

Round 1

Reviewer 1 Report

Overall the systematic review provides a snapshot of the literature that captures the expression profiles of selected non-coding mRNAs in SCM and FF. The review is of value to those in the field studying potential non-invasive markers of embryo quality. However, there are some concerns and edits that should be addressed.

First, the number of papers used in the analysis is fairly small at only 18. 

Second, the Intro is missing some relevant background in this field. I suggest adding info about niPGT-A as this technique used SCM for embryo ploidy status determination. Line 79 of the Intro ("short time frame") needs to be expanded as that is too vague of a description, especially considering that IVF clinics rarely have the ability to detect non-coding RNAs in house.

Third, Line 44 of the Intro is very confusing with the statement "embryo or blastocyst is replaced"...what does "replaced" refer to? Uterine transfer?

Another limitation is the lack of discussion of how the non-coding RNAs relate to embryo ploidy status. As PGT-A is fairly common, analyzing SCM could be considered an additional metric when selecting an embryo for transfer. The authors should review the data and see if there is any correlation with embryo ploidy status and specific non-coding RNA expression.

Author Response

Thanks for your decision letter and for the reviewers’ comments concerning our manuscript [No: ijms-2248005] entitled “Non-coding RNAs as biomarkers for embryo quality and pregnancy outcomes: A systematic review and meta-analysis”. Those comments are valuable and helpful for revising and improving our manuscript. We have studied the comments carefully and have made some changes to the manuscript, which we hope to meet with approval. Revised portions are shown in track changes in the revised manuscript. The main corrections in the revised manuscript and the point-by-point responses to Reviewer 1 comments are attached. Please see the attachment.

For the Editor's comment: Please kindly rearrange Materials and Methods section to the section before Results as per the journal's requirement for systematic review. The structure should be 1. introduction 2. methods 3. results and discussion 4. conclusions.

Response: We have rearranged the Materials and Methods section accordingly (Lines 95-154).

Reviewer 2 Report

Despite advances in in vitro fertilization (IVF), there is still a lack of non-invasive and reliable biomarkers for selecting embryos with the highest developmental and implantation potential.  sncRNAs have been identified in biological fluids, and extracellular sncRNAs are explored as diagnostic biomarkers in the prediction of IVF outcomes. To comprehensively understand the potential of using sncRNAs as non-invasive biomarkers in embryo assessment, authors systematically reviewed studies investigating the correlation of sncRNAs levels in various extracellular environments with pregnancy outcomes or embryo quality parameters. 

The review was well organized, and comments were listed as below.

1. The references should be updated.

2. The English of the manuscript should be improved by native English speakers to make it more readable.

3. There are so many sncRNA candidates, could you please furtherly narrow them?

Author Response

Thanks for your decision letter and for the reviewers’ comments concerning our manuscript [No: ijms-2248005] entitled “Non-coding RNAs as biomarkers for embryo quality and pregnancy outcomes: A systematic review and meta-analysis”. Those comments are valuable and helpful for revising and improving our manuscript. We have studied the comments carefully and have made some changes to the manuscript, which we hope to meet with approval. Revised portions are shown in track changes in the revised manuscript. The main corrections in the revised manuscript and the point-by-point responses to Reviewer 2 comments are attached. Please see the attachment.

For the Editor's comment: Please kindly rearrange Materials and Methods section to the section before Results as per the journal's requirement for systematic review. The structure should be 1. introduction 2. methods 3. results and discussion 4. conclusions.

Response: We have rearranged the Materials and Methods section accordingly (Lines 95-154).

Reviewer 3 Report

The authors performed a systematic review and meta-analysis to determine the predictive role of sncRNAs in embryo quality and IVF outcomes. They demonstrated that sncRNAs might distinguish embryos with higher developmental and implantation potentials. 

The study design is accurate and the manuscript is well written. The results are described with appropriate predictive power and clear tables. The findings are described according to the current literature and provide more convincing evidence of the role of sncRNAs in the physiology of human embryo development

Please refer to attachment for additional comments.

Author Response

Thanks for your decision letter and for the reviewers’ comments concerning our manuscript [No: ijms-2248005] entitled “Non-coding RNAs as biomarkers for embryo quality and pregnancy outcomes: A systematic review and meta-analysis”. Those comments are valuable and helpful for revising and improving our manuscript. We have studied the comments carefully and have made some changes to the manuscript, which we hope to meet with approval. Revised portions are shown in track changes in the revised manuscript. The main corrections in the revised manuscript and the point-by-point responses to Reviewer 3 comments are attached. Please see the attachment.

For the Editor's comment: Please kindly rearrange Materials and Methods section to the section before Results as per the journal's requirement for systematic review. The structure should be 1. introduction 2. methods 3. results and discussion 4. conclusions.

Response: We have rearranged the Materials and Methods section accordingly (Lines 95-154).

Round 2

Reviewer 1 Report

Edits are acceptable.

Reviewer 2 Report

Comments have been addressed. Acception is recommened.